# The Mediating Role of Work–Life Balance Between Perceived Partner Support and Satisfaction with Food-Related Life in Dual-Earning Parents and Their Adolescents

**DOI:** 10.3390/nu17061018

**Published:** 2025-03-14

**Authors:** Berta Schnettler, Andrés Concha-Salgado, Klaus G. Grunert, Ligia Orellana, Mahia Saracostti, Katherine Beroiza, Héctor Poblete, Germán Lobos, Cristian Adasme-Berríos, María Lapo, Leonor Riquelme-Segura, José A. Sepúlveda, Karol Reutter, Enid Thomas

**Affiliations:** 1Facultad de Ciencias Agropecuarias y Medioambiente, Universidad de La Frontera, Temuco 4811230, Chile; 2Centro de Excelencia en Psicología Económica y del Consumo, Universidad de La Frontera, Temuco 4811230, Chile; ligia.orellana@ufrontera.cl (L.O.); k.beroiza01@ufromail.cl (K.B.); hector.poblete@ufrontera.cl (H.P.); jose.sepulveda@ufrontera.cl (J.A.S.); k.reutter01@ufromail.cl (K.R.); 3Scientific and Technological Bioresource Nucleus (BIOREN-UFRO), Universidad de La Frontera, Temuco 4811230, Chile; 4Facultad de Especialidades Empresariales, Universidad Católica de Santiago de Guayaquil, Guayaquil P.O. Box 09-01-4671, Ecuador; maria.lapo@cu.ucsg.edu.ec; 5Departamento de Psicología, Universidad de La Frontera, Temuco 4811230, Chile; andres.concha@ufrontera.cl; 6MAPP Centre, Aarhus University, 8000 Aarhus, Denmark; klg@mgmt.au.dk; 7Departamento de Trabajo Social, Universidad de Chile, Santiago 7800284, Chile; mahia.saracostti@uchile.cl; 8Facultad de Economía y Negocios, Universidad de Talca, Talca 3465548, Chile; globos@utalca.cl; 9Departamento de Economía y Administración, Universidad Católica del Maule, Talca 3530000, Chile; cadasme@ucm.cl; 10Departamento de Trabajo Social, Universidad de La Frontera, Temuco 4811230, Chile; leonor.riquelme@ufrontera.cl; 11Facultad de Ciencias Agropecuarias y Medioambiente, Doctorado en Ciencias Agroalimentarias y Medioambiente, Universidad de La Frontera, Temuco 4811230, Chile; e.thomas01@ufromail.cl

**Keywords:** partner support, work–life balance, well-being, dual-earning parents, adolescents

## Abstract

Background: Partner support and work–life balance (WLB) are important for families’ well-being. Previous research has linked perceived family support, WLB, and satisfaction with food-related life (SWFoL); however, there is limited information regarding each parent’s support from their partner. Drawing on the conservation of resources theory, the work–home resources, and the actor–partner interdependence model (APIM), this study examined the direct and indirect effects of parents’ perceived partner support, WLB, and the SWFoL of dual-income parents and their adolescents, considering the moderating role of parents’ gender-transcendent attitudes. Methods: A total of 516 dual-earning parents with one adolescent child were recruited in Chile using non-probability sampling. The parents completed scales measuring perceived partner support, WLB, and gender role attitudes. The three family members responded to the Satisfaction with Food-Related Life scale. The data were analyzed using the mediation APIM, structural equation modeling, and multi-group analysis. Results: The model fit was robust (RMSEA = 0.016; SRMR = 0.052; CFI = 0.982; TLI = 0.978). The findings showed that the mothers’ SWFoL was indirectly and positively affected by their and the fathers’ perceived partner support through both parents’ WLB. The fathers’ SWFoL was directly and positively affected by their perceived partner support and indirectly via their WLB. The adolescents’ SWFoL was directly and positively affected by the mothers’ perceived partner support and indirectly by the fathers’ perceived partner support through the fathers’ WLB. In families where the fathers exhibited low gender-transcendent attitudes, the relationship between the mothers’ perceived partner support and WLB was stronger. Conclusions: Thus, it can be concluded that the mediating role of work–life balance is significant, as it facilitates the transmission of resources within and between individuals to enhance parents’ and adolescents’ SWFoL.

## 1. Introduction

Satisfaction with food-related life (SWFoL) pertains to how people assess their food selections and eating practices, encompassing meal planning, grocery shopping, food preparation, eating habits, and food waste management [1]. SWFoL matters extend beyond just nutritional concerns; they also reflect the social aspects of food consumption. For example, family meals are a significant opportunity for family members to engage with each other and share personal resources, such as emotional and instrumental support between parents and their children [2,3]. SWFoL is therefore affected not only by directly food-related activities, like preparing and serving meals, but also by how the interaction of family members creates conditions in the home conducive to creating enjoyable meals for all family members.

In explaining how family interaction affects SWFoL, the conservation of resources theory (COR) [4] can be involved. This theory addresses how social support—the emotional or instrumental resources obtained through social relationships that help alleviate stress and improve well-being [5]—can foster a better family life, including the food-related life. Social support can come from various sources, such as general social networks, family assistance, and partner or spousal support [6]. Emotional partner support manifests in how well one partner understands and empathizes with the other’s problems. In contrast, instrumental partner support refers to tangible help, such as preparing meals. A partner’s housework often gauges the extent of instrumental support. In contrast, emotional support is typically assessed through a partner’s empathy or emotional engagement with personal matters. Insufficient or ineffective support can lead to higher levels of stress and strain in the future [7]. Our research will specifically examine the impact of partners’ emotional and instrumental support on SWFoL.

Social support can have numerous other effects on the family, and it can have effects beyond the boundaries of the household, i.e., in family members’ job lives and how this, in turn, relates to the conditions of family living. Numerous studies have investigated the concept of work–life balance (WLB) [7]. WLB adopts a broad perspective regarding the interaction between an employee’s work responsibilities and roles, encapsulated in the definition as “the extent to which an individual can adequately manage the multiple roles in their life, including work, family, and other major responsibilities” [8]. When both partners experience low WLB, it can negatively impact family well-being, relationship quality, and marital satisfaction. This, in turn, may lead to reduced family cohesion and adversely affect the well-being of children [9]. In contrast, evidence indicates that employees who perceive a balanced integration of their work and personal life will likely experience fewer health issues, enhanced well-being, and increased satisfaction levels across various domains, including job, family, and food-related life [10]. It is natural to expect that WLB can also affect SWFoL.

Although a husband or partner is often viewed as a crucial source of support, our understanding of how spousal support operates within dual-career families is still limited. Only a few studies have explored this issue [11], and even fewer have examined the relationship between partner support, WLB, and SWFoL within this family structure. Previous research in Chile has looked into some of these connections in isolation. For instance, Schnettler et al. [12] investigated how perceived emotional family support and parents’ WLB influence adolescents’ SWFoL. Another study [10] explored the relationship between WLB and SWFoL among dual-income couples. Additionally, Schnettler et al. [13] examined the connections between parents’ emotional family support, WLB, and SWFoL in families with adolescent children. These studies demonstrated crossover effects among various family members. Crossover denotes sharing resources between individuals within a dyad. Crossover effects can be symmetrical, meaning that each partner influences the other (for example, a husband affecting his wife and vice versa). However, they can also be asymmetrical, where only one partner influences the other, without a reciprocal effect (for instance, a husband influencing his wife but not the wife influencing her husband) [14]. Achieving a successful equilibrium between work and other aspects of life can serve as both a precursor to and a result of how individuals effectively manage their resources [15]. Maintaining a balance between work and other aspects of life is not just an individual duty; it frequently involves the people surrounding them [14]. This flow of resources plays a crucial role in influencing an individual’s well-being and intimate relationships [8], underscoring the interconnected nature of family dynamics.

The relationship between partner support, WLB, and SWFoL can be effectively analyzed through the work–home resources (W–HR) model established by ten Brummelhuis and Bakker [16], as well as through the lens of the COR theory, which emphasizes how resource accumulation fosters further benefits [17]. According to COR theory, WLB can be conceptualized as a dynamic process whereby resources derived from the home environment, particularly perceived partner support, contribute to forming a “gain spiral.” This positive feedback loop can increase satisfaction with various aspects of family life, including enhanced levels of SWFoL [16]. The insights from this research are vital for comprehending the intricate interplay between partner support, WLB, and SWFoL.

Utilizing the COR theory and the W–HR model, this study investigates the direct and indirect effects of perceived partner support, work–life balance, and satisfaction with food-related life among parents and one adolescent child in dual-income families in Chile. Our analytical approach employs the mediation actor–partner interdependence model proposed by Kenny et al. [18] and Ledermann et al. [19]. This model proficiently captures individual (actor effects) and interdependent influences from partners (partner effects or crossover) on the outcome variables. In this framework, dyadic interactions—such as those between mothers and fathers, mothers and adolescents, and fathers and adolescents—are considered the unit of analysis [18]. Actor effects refer to the relationships between each parent’s perceived partner support, their own WLB and SWFoL, and the connection between their WLB and SWFoL. On the other hand, partner effects examine how one parent’s perceived partner support influences the WLB and SWFoL of the other parent, including how one parent’s WLB connects to the other parent’s SWFoL. In this research, adolescents are solely considered as partners who are impacted by their parents’ perceived partner support and WLB regarding their SWFoL.

Lastly, the division of housework is linked to attitudes toward gender roles [20,21], which reflect how much individuals endorse traditional gender roles [22]. Additionally, women are often primarily responsible for household tasks and caregiving, even when employed. Meanwhile, their husbands’ support may be minimal, as many men perceive their primary role as providing financial support and protecting the family [11]. Consequently, it can be anticipated that those with more traditional views on gender roles may experience notable differences in the relationships between partner support, WLB, and SWFoL compared to those who uphold more egalitarian or gender-transcendent views. Based on this, the study explored how parents’ gender-transcendent attitudes might be a moderating factor.

### 1.1. Conceptual Framework and Hypotheses Development

We argue that satisfaction with food-related life depends on the availability and deployment of family resources, and that the work–life balance of the family members partly mediates this process. Those with a more extensive reservoir of resources are better positioned to generate further resource gains, leading to what are known as gain spirals. Interpersonal resources can significantly aid in allocating personal resources [14]. For instance, support from a spouse can improve the partner’s ability to effectively allocate personal resources for both work and non-work needs [23].

### 1.2. Partner Support, Work–Life Balance, and Satisfaction with Food-Related Life

The W–HR model suggests that resources from the home environment can enhance performance in various areas of life [16]. This model suggests that resources from the home environment can contribute to the growth of personal resources, which can be utilized in the food environment. Although, to the authors’ knowledge, there are no available studies that have assessed the relationship between partner support and SWFoL, a positive and direct association between perceived emotional family support and SWFoL has been reported in fathers, but not mothers, in dual-income parents [13]. This finding suggests that family emotional support can be a resource that may help family members achieve better performance when preparing and serving meals. However, in their qualitative study involving working women in Saudi Arabia, Alarifi and Basahal [24] discovered that for women, instrumental assistance with household tasks, such as cooking and childcare/parenting duties, is one of the partners’ most significant and appreciated forms of support.

Recent studies have indicated a growing trend in which an increasing number of fathers assume significant household responsibilities to alleviate the stress experienced by working mothers. This shift allows them to provide practical and emotional support to their wives [25]. Although Schnettler et al. [13] did not find crossover effects between mothers’ and fathers’ family emotional support and SWFoL, we argue that considering both emotional and instrumental partner support may allow parents to transfer resources from one parent to the other via crossover. Regarding adolescents, emotional family support positively correlates with SWFoL in Chilean adolescents [12].

The COR theory posits a positive relationship between partner support and achieving a healthy WLB [26]. Previous studies in Chile found that family emotional support improves WLB in fathers and mothers [27,28]. Furthermore, Jeong et al. [29] found that in Korea, perceived spousal support positively affects WLB for wives and husbands.

Regarding crossover, in previous studies with Chilean couples, fathers’ perceived family support positively crossed over to the mothers, improving their WLB, but not vice versa [27,28]. Similarly, in Korea, Jeong et al. [29] found that only the perceived support from the husband positively impacted the wife’s WLB, but not vice versa.

Research indicates that employees struggling to balance work and home responsibilities often create less healthy living environments [30], participate in fewer family meals, and consume fast food more frequently [31]. Collectively, these factors not only harm the dietary quality of both partners [32] but can also be expected to detract from the social and emotional qualities of family meals.

Research grounded in the W–HR model suggests that achieving a WLB can benefit employees and their families, particularly within the food domain [16,33]. Evidence indicates that women who effectively juggle work and home responsibilities are more inclined to enjoy regular family meals and offer healthier food choices [34]. These beneficial outcomes are subsequently associated with enhanced SWFoL among adults and adolescents (e.g., [3,35,36]). In this regard, an earlier study in Chile reported that the fathers’ WLB crosses over to the mothers’, positively affecting their SWFoL [13]. Additionally, the WLB of both mothers and fathers is positively associated with their children’s SWFoL [12].

### 1.3. The Mediating Role of Work–Life Balance

In addition to the direct impacts of emotional and instrumental support, both forms can also lead to indirect effects. In this regard, at an individual level, Russo et al. [37] found that WLB mediates the relationship between family support and the employee’s positive energy at work. More recently, it was reported that WLB mediates the relationship between emotional family support and life satisfaction in parents of dual-income families [27]. Similarly, Nabawanuka and Ekmekcioglu [38] reported that WLB is a mediator between perceived supervisor support and the employee well-being of millennial employees. Furthermore, Schnettler et al. [13] found that WLB mediates the relationship between emotional family support and SWFoL in dual-income families, irrespective of the parent’s gender. It should be noted that these authors did not assess the potential mediating role of WLB between parents’ emotional family support and their adolescent children’s SWFoL. Nevertheless, we argue that this positive relationship may also go beyond extending to children. Consequently, WLB could also act as a mediator at the interindividual level.

### 1.4. The Impact of Parental Attitudes That Go Beyond Conventional Gender Roles

Gender role attitudes are crucial in shaping individuals’ identities, behaviors, and responsibilities [39], and they have significant consequences. People with low gender-transcendent attitudes typically view a woman’s primary responsibility as being in the home, prioritizing family duties. Conversely, those with high gender-transcendent attitudes or egalitarian views promote the idea of equal responsibilities for both men and women [40].

Women who identify as egalitarian, who advocate for equal sharing of household duties, typically do less housework compared to traditionalists. The pattern is similar for men but reversed: traditional men are less involved in household tasks, while egalitarian men assume more responsibilities [20]. Despite this, studies reveal an ongoing gender gap in housework contributions [41]. For instance, previous studies involving adults in Switzerland and Canada showed that women in most age groups reported cooking more often at home than did men [21,41].

### 1.5. Hypotheses

Based on this background, we propose the following hypotheses (Figure 1):

**H1.** 
*Perceived partner support is positively related to satisfaction with food-related life for fathers and mothers.*


**H2.** *One parent’s perceived partner support is positively linked to (a) the other parent’s and (b) adolescents’ satisfaction with food-related life*.

**H3.** 
*Perceived partner support positively relates to work–life balance for fathers and mothers.*


**H4.** 
*One parent’s perceived partner support positively relates to the other parent’s work–life balance.*


**H5.** *Work–life balance is positively associated with satisfaction with food-related life for fathers and mothers*.

**H6.** *One parent’s work–life balance is positively associated with (a) the other parent’s and (b) adolescents’ satisfaction with food-related life*.

**H7.** 
*Work–life balance mediates the relationship between parents’ perceived partner support and the three family members’ satisfaction with food-related life (actor and partner effects).*


**H8.** 
*Gender-transcendent attitudes of mothers and fathers influence the relationships between perceived partner support, work–life balance, and satisfaction with food-related life.*


This research was conducted in Chile, a developing country in South America. According to data from the last census in 2017 [42], the country’s population reached 17,574,003 inhabitants. Over the years, the population has aged, which is evident in the changes in the distribution of age groups: there is a decrease in individuals aged 0–14 and an increase in those 65 and older. Most of the population lives in urban areas, with only 12.2% residing in rural regions. Approximately 70% of the population in various regions of Chile works in the tertiary sector. The average years of schooling for those aged 25 is 11.05. The average household size is 3.1 people. Two-parent nuclear households with children represent 28.8%, with an increase in single-person households compared to the number in the 2002 census (17.8%). A man heads 58.4% of all households, while a woman heads 41.6%. Most of the population falls into the middle (52%) and lower (31%) socioeconomic levels.

Chilean households exhibit unhealthy eating patterns, characterized by high expenditures on sugary drinks and sweets, alongside an insufficient consumption of fruits, vegetables, fish, and legumes to meet the recommendations for a healthy diet. This issue is particularly pronounced in dual-earning families. Evidence indicates that unhealthy eating habits are prevalent among employees across various economic sectors, negatively affecting their quality of life. Despite these challenges, previous studies show that members of dual-earning families report relatively high levels of SWFoL, averaging around 21 on a scale of 30 [28,33].

## 2. Materials and Methods

### 2.1. Sample and Procedure

In this study, 516 dual-earning families were selected using a non-probabilistic convenience sampling approach, employing quotas that reflect the distribution of families across socioeconomic levels (high, medium, and low) in Temuco, Chile. This method was developed to guarantee a diverse sample representing various socioeconomic backgrounds. The sample size we gathered is consistent with the recommendations from Ledermann et al. [43], who suggest that for adequately identifying mediated pathways among distinguishable dyads, a minimum of 91 dyads is needed, while 249 dyads are required to evaluate actor and partner effects. Our sample size surpasses these recommendations, as we strive to represent the diversity of Chilean families. Each family included a mother, a father, and one adolescent aged between 10 and 15, all residing in Temuco, Chile. Families were reached through the teenagers’ schools and social networks. Families contacted via their children’s schools received a letter of invitation with information about the study. If a family had more than one teenage child, the child contacted through the schools responded. If contacted through social media, the family chose which child responded. All three family members who agreed to participate were assigned an interviewer. Trained interviewers outlined the study’s goals and the questionnaire format to the parents, ensuring that their responses remained confidential and anonymous. Families interested in participating provided an email address, which was used to send survey links to all three family members, with instructions for each family member to answer the questionnaires separately. Interviewers also offered phone support to answer any questions and facilitate the completion of the questionnaires. Data collection took place from June to November 2023.

At the start of the online questionnaire, mothers and fathers received consent forms, while adolescents were presented with an assent form. Parents and adolescents indicated their willingness to participate by checking a box. The questionnaires were stored separately in three databases on the QuestionPro platform (QuestionPro Inc., Seattle, WA, USA). After completing the three questionnaires, the families were compensated via bank transfer (USD 15).

A pilot test comprising 40 families was conducted using the same recruitment strategy and data collection process, which required no modifications. This research is part of a more extensive study exploring the connections between work, family, food demands and resources, and well-being in Chilean households. The Universidad de La Frontera Ethics Committee approved this study (protocol number 035-23).

### 2.2. Measures

Mothers and fathers responded to the following scales:

Perceived partner support [7] refers to both emotional and instrumental support from partners, which is evaluated using three items forming a single dimension. One such item is, “Partner lets me know that he/she understands me”. The study employed the Spanish version of the Perceived Partner Support scale [28]. Participants rated each statement on a 5-point scale, from never (1) to always (5). The perceived partner support scores were derived by summing the ratings of the three items. Higher perceived partner support scores mean that one partner received more emotional and instrumental support from the other.

Work–life balance (WLB) [8] was measured using a scale consisting of three items that form a single dimension (for example, “I manage to balance the demands of my work and personal/family life well”). This study utilized the validated Spanish version of the WLB scale [12], demonstrating good internal consistency in samples of Chilean adults and adolescents [10,12,13]. Participants were asked to express their agreement level with the three statements using a 5-point Likert scale (1: completely disagree; 5: completely agree). The WLB scores were calculated by adding the scores from the three items. Higher WLB scores reflect a greater work–life balance.

The Gender Role Attitudes Scale (GRAS) [44] includes two subscales: transcendent attitudes, which have 5 items, and stereotype attitudes, consisting of 15 items. These subscales assess gender role attitudes within key socialization areas, such as family, social circles, and the workplace. Our research focused exclusively on the transcendent attitudes dimension, which measures beliefs regarding roles that transcend traditional gender norms (e.g., “Men have the same responsibilities for household chores as women”). Participants evaluated each statement using a 5-point Likert scale (1: totally disagree; 5: totally agree). We utilized the version of the scale validated in Chile by Pérez et al. [45], which reported an omega coefficient of 0.79 for the transcendent attitudes dimension. Higher scores reflect greater support for gender role egalitarianism, while lower scores indicate less support.

Mothers, fathers, and adolescents answered using the following scale:

Satisfaction with food-related life (SWFoL) refers to a scale [1] composed of five items that assess an individual’s overall evaluation of their food and eating habits (for instance, “Food and meals are very positive elements in my life”). The Spanish-validated version of SWFoL [46] scale was utilized, demonstrating good internal reliability in samples of adults and adolescents [10,12,13,33]. Participants were asked to express their agreement level with each statement using a 6-point Likert scale (1: completely disagree; 6: completely agree). SWFoL scores were calculated by summing the scores from the five items. Higher SWFoL scores reflect greater satisfaction with food-related life.

The three family members were asked about their ages. Adolescents were also asked about their gender, while parents provided information regarding their employment status, weekly work hours, work arrangements (remote, in-person, or hybrid), and the number of hours per day during a week they spend on childcare, housework, and cooking. Furthermore, women were surveyed about their family size, the number of children, and how many days each week the family shares meals (including breakfast, lunch, supper, and dinner). The family’s socioeconomic status (SES) was assessed based on combined income, household size, and the education and occupation of the partner with the highest income [47].

### 2.3. Data Analysis

Descriptive analyses were conducted using SPSS v. 23. The statistical comparison of perceived partner support and WLB between mothers and fathers was performed using the Wilcoxon signed-rank non-parametric test (due to the violation of normality). Furthermore, the comparison among the three family members’ SWFoL was analyzed through repeated measures analysis of variance (RM ANOVA) and specific post hoc comparisons with the Bonferroni correction.

The Perceived Partner Support scale had not previously been utilized in dyadic analysis. To address this gap, a dyadic confirmatory factor analysis (CFA) was implemented, following the methodology established by Claxton et al. [48], to investigate its latent structure and psychometric characteristics. Internal consistency was evaluated with the omega coefficient [49]. Convergent validity was assessed by analyzing the standardized factor loadings of the scale (preferably over 0.5), along with their statistical significance and the average variance extracted (AVE, with values exceeding 0.5) [49].

To investigate how perceived partner support affects WLB and SWFoL, we employed the mediation actor–partner interdependence model (APIM) with distinguishable dyads, utilizing structural equation modeling (SEM) with latent variables [18].

The APIM controls the mutual impact of the independent variable of both parents by looking at the correlations among their perceived partner support. Furthermore, it considers other sources of interdependence by analyzing the correlations between the residual errors of each member’s dependent variables, specifically WLB between parents and SWFoL between the three family members [18].

To guarantee a precise alignment of the data, the analysis includes variables that directly influence the outcomes for parents, i.e., WLB and the three family members’ SWFoL. The model explicitly considers the ages of the three family members, the total number of children, types of employment, hours worked, family SES, and the frequency of shared suppers among all family members each week. Previous studies have indicated that the significance placed on health-related aspects of SWFoL increases as individuals age [50], prompting the inclusion of parents’ age as a control variable. The age of adolescents was also considered, as research suggests that older adolescents tend to better understand how their parents’ jobs impact their lives compared to younger ones [51]. The number of children was included because a previous study reported that this variable negatively influenced their parents’ WLB [13]. The type of employment was factored in, given that self-employed respondents experience greater WLB, which has been associated with higher involvement in food-related tasks that increase SWFoL [10]. Working hours were also considered, since part-time workers often have lower monthly incomes than full-time employees, which can influence SWFoL [28]. Family SES was included as another control variable, as employees from lower SES backgrounds tend to report decreased SWFoL [13]. Lastly, the frequency of shared suppers among all family members each week was included because it improves SWFoL in adults and adolescents [33].

The statistical software Mplus 8.11 was used to perform CFA and SEM. The unweighted least squares mean and variance adjusted (ULSMV) method was employed to estimate the factor loadings and the structural model parameters. The items were included on an ordinal scale, so the CFA and SEM analyses were conducted using the polychoric correlation matrix. The adequacy of the model was assessed using the Tucker–Lewis index (TLI), the comparative fit index (CFI), the root mean square error of approximation (RMSEA), and the standardized root mean square residual (SRMR). Specifically, TLI and CFI values greater than 0.95 indicate a good fit. RMSEA values below 0.06 and SRMR values below 0.08 suggest a good fit [52].

We conducted an SEM analysis with a bias-corrected (BC) bootstrap confidence interval using 1000 samples to assess the mediating effects of WLB, which aligns with the methodology described by Lau and Cheung [53]. Support for a mediating effect was identified when the BC confidence interval did not include zero.

Additionally, we investigated the moderating effects proposed in our research question through multi-group analysis, as Ryu and Cheong [54] outlined. This involved comparing direct effect parameters across groups for each model path, as determined by the dichotomous moderators. Evidence of a moderation effect was identified when a statistically significant difference in a direct estimate was found between groups within the model.

## 3. Results

### 3.1. Sample Description

The study included 516 dual-earner families with adolescents aged between 10 and 15, comprising 51.0% males and 49.0% females. This resulted in responses from 516 mothers, 516 fathers, and 516 adolescents, amounting to 1548 participants. The sociodemographic characteristics are detailed in Table 1.

Table 2 presents the average scores and relationships between the support perceived by mothers and fathers from their partners, WLB, and the three family members’ SWFoL. All the correlations were statistically significant and aligned with the expected patterns. According to the non-parametric Wilcoxon comparison, fathers reported significantly higher levels of perceived partner support than mothers: W = 17,597.5, *p* < 0.001, r_matched rank biserial_ = −0.364 (medium effect). No significant differences were found in the mother–father WLB comparison: W = 30,451, *p* = 0.341, r_matched rank biserial_ = −0.058. Fulfilling the assumption of sphericity, the repeated measures analysis of variance test showed significant overall differences of small magnitude when comparing the total SWFoL means, according to the participant’s role in the family (mother, father, or child): F(2, 1030) = 24.999, *p* < 0.001, partial *η^2^* = 0.046. Specific post hoc tests with Bonferroni correction found that adolescents reported significantly higher SWFoL than did their parents (small effects), and that the latter did not differ from each other.

### 3.2. Psychometric Properties of the Perceived Partner Support Scale

The dyadic CFA findings showed that the Perceived Partner Support Scale fits the data well for mothers and fathers (RMSEA = 0.047; SRMR = 0.052; CFI = 0.998; TLI = 0.995). The scale exhibited high reliability, with omega coefficients of 0.94 for mothers and 0.93 for fathers. Additionally, all factor loadings were statistically significant (*p* < 0.001), and their values indicated strong convergent validity (ranging from 0.901 to 0.919 for mothers and from 0.864 to 0.961 for fathers). The AVE was computed at 0.83 for mothers and 0.82 for fathers.

### 3.3. Investigating Actor–Partner Hypotheses

The factor loadings for Perceived Partner Support, Work–Life Balance, and Satisfaction with Food-Related Life scales exceeded 0.50 and were statistically significant (*p* < 0.001). Moreover, the AVE values were above 0.50, while the omega coefficients indicated high reliability across all measures (Table 2).

Figure 2 illustrates the mediation APIM analysis, emphasizing the framework’s direct associations. This model assessed the interrelationship between parents’ perceived partner support and WLB, alongside the SWFoL reported by the three family members. The model demonstrated a robust fit with the dataset, indicated by fit indices including CFI = 0.982; TLI = 0.978; and RMSEA = 0.016. A statistically significant correlation was identified between the perceived partner support of mothers and fathers (r = 0.546, *p* < 0.001). Additionally, significant correlations were found between the residual errors of parents’ WLB (r = 0.349, *p* < 0.001) as well as between the SWFoL of mothers and fathers (r = 0.411, *p* < 0.001). Correlations were also noted between mothers’ and adolescents’ SWFoL (r = 0.375, *p* < 0.001) and between fathers’ and adolescents’ SWFoL (r = 0.386, *p* < 0.001).

The analysis indicates that most control variables did not significantly affect the model. Notably, the number of children negatively influenced mothers’ WLB (γ = −0.101, *p* = 0.033) and SWFoL (γ = −0.127, *p* = 0.008). Additionally, mothers’ increased working hours per week detrimentally impacted their WLB (γ = −0.122, *p* = 0.021). In contrast, the type of employment positively influenced mothers’ WLB, as self-employed mothers reported higher levels of WLB than those employed in traditional roles (γ = −0.107, *p* = 0.019). Furthermore, the frequency of family suppers shared among all members positively contributed to mothers’ WLB (γ = 0.207, *p* < 0.001) and SWFoL (γ = 0.148, *p* = 0.001), as well as adolescents’ SWFoL (γ = 0.130, *p* = 0.006).

Based on the data analysis presented in Figure 2, a positive association was identified between fathers’ perceived partner support and their SWFoL, with a coefficient of γ = 0.175 (*p* = 0.002). In contrast, mothers’ perceived partner support did not exhibit a significant relationship with their SWFoL, as evidenced by γ = 0.067 (*p* = 0.213). These results affirm H1 for fathers, while failing to do so for mothers. Additionally, it was noted that mothers’ perceived partner support was positively associated with adolescents’ SWFoL (γ = 0.122, *p* = 0.035). However, this association was insignificant for fathers’ SWFoL, which recorded a coefficient of γ = 0.017 (*p* = 0.769). Furthermore, fathers’ perceived partner support was not significantly related to mothers’ SWFoL (γ = 0.068, *p* = 0.230) or adolescents’ SWFoL (γ = 0.097, *p* = 0.085). These outcomes do not support hypothesis H2a but support hypothesis H2b regarding mothers.

The findings reveal a significant positive association between perceived partner support and WLB for both mothers (γ = 0.291, *p* < 0.001) and fathers (γ = 0.344, *p* < 0.001). These outcomes support H3 for both parental figures. Additionally, the perceived partner support reported by fathers was positively associated with the WLB of mothers (γ = 0.159, *p* = 0.002). Conversely, the mothers’ perceived partner support did not show a statistically significant relationship with fathers’ WLB (γ = 0.110, *p* = 0.064), thus validating H4 exclusively for fathers.

The findings of the study indicated a significant positive relationship between WLB and SWFoL for both mothers (γ = 0.409, *p* < 0.001) and fathers (γ = 0.146, *p* = 0.005), thereby confirming H5 for both parental groups. Additionally, it was observed that the WLB of fathers exhibited a positive relationship with the SWFoL of mothers (γ = 0.232, *p* < 0.001). In contrast, the WLB of mothers did not demonstrate a significant association with the SWFoL of fathers (γ = 0.016, *p* = 0.769), indicating that Hypothesis 6a was upheld only for fathers. Furthermore, while fathers’ WLB was positively associated with adolescents’ SWFoL (γ = 0.016, *p* = 0.769), mothers’ WLB did not show a significant relationship with adolescents’ SWFoL (γ = 0.212, *p* < 0.001), thereby supporting Hypothesis 6b exclusively for fathers.

Straight black bold solid-lined arrows with one head represent significant direct effects. Straight gray dotted arrows represent nonsignificant direct effects. Curved double-headed arrows represent the correlations between fathers’ and mothers’ perceived partner support and the correlations between the residual errors of each of the three family members’ SWFoL. Direct results were obtained through the APIM analysis.

The path diagram does not display the indirect effects of WLB (H7), the moderating role of gender-transcendent attitudes (H8), or account for the effects of both partners’ age, employment type, working hours, family SES, number of children, and frequency of family suppers per week on the dependent variables (WLB and SWFoL).

### 3.4. Testing the Mediating Role of Work–Life Balance

The study revealed that mothers’ WLB mediates the relationship between their perceived partner support and SWFoL (Table 3). This finding is substantiated by a significant indirect effect (standardized indirect effect = 0.043, 95% CI = 0.009, 0.076, *p* = 0.013). Similarly, fathers’ WLB was confirmed to mediate the association between perceived partner support and SWFoL, reflected in a statistically significant indirect effect (standardized indirect effect = 0.141; 95% CI = 0.081, 0.200; *p* < 0.001).

Moreover, fathers’ WLB was evident as a mediator in the relationship between their perceived partner support and the SWFoL of both mothers (standardized indirect effect = 0.080; 95% CI = 0.038, 0.122; *p* < 0.001) and adolescents (standardized indirect effect = 0.073; 95% CI = 0.032, 0.113; *p* < 0.001). Mothers’ WLB mediated the relationship between fathers’ perceived partner support and mothers’ SWFoL (standardized indirect effect = 0.023; 95% CI = 0.038, 0.122; *p* = 0.035). Nonetheless, no additional mediating roles of WLB were identified, thus providing partial support for hypothesis H7.

### 3.5. The Impact of Parental Attitudes That Go Beyond Conventional Gender Roles

The study explored parents’ gender-transcendent attitudes through multi-group analyses (Hypothesis 8), treating these attitudes as a categorical variable based on the median scores of the attitudes measured for mothers (median = 28) and fathers (median = 25). This allowed for the comparison of low versus high gender-transcendent attitudes for each parent.

The results of the multi-group analysis for mothers indicated a strong alignment between the fit indices and the data (RMSEA = 0.041; CFI = 0.964; TLI = 0.963). However, mothers’ gender-transcendent attitudes did not moderate any of the relationships posited in Hypotheses 1–6. This result did not support H8 for mothers.

The findings from the multi-group analysis for fathers demonstrated good alignment between the fit indices and the data (RMSEA = 0.035; CFI = 0.972; TLI = 0.972). It was noted that fathers’ attitudes toward gender transcendence influenced the relationship between mothers’ perceived partner support and WLB (γ = 0.276, *p* = 0.045). This relationship was more robust in families where fathers had low gender-transcendent attitudes (γ = 0.441, *p* < 0.001) and weaker in families with fathers who had high gender-transcendent attitudes (γ = 0.248, *p* < 0.001). This result partially supported H8 for fathers.

## 4. Discussion

Research has established a connection between perceived family support, work–life balance (WLB), and satisfaction with food-related life (SWFoL) among dual-earning parents with adolescents. However, there is limited information about the support each parent receives from their partner. This study is the first to comprehensively examine the direct and indirect effects of perceived partner support, WLB, and SWFoL in dual-income families with adolescents. Additionally, it explores the moderating role of parents’ gender-transcendent attitudes. This research offers new insights into the crossover effects and indirect effects primarily stemming from the support that fathers receive from mothers.

Our research, which employed mediation APIM and structural equation modeling, has produced findings with considerable practical significance. We identified that fathers’ perceptions of partner support are directly and positively associated with their SWFoL. Additionally, fathers’ perceived partner support also indirectly affects the well-being of mothers and adolescents via WLB. On the other hand, we observed that mothers’ perceived partner support directly and positively influences the SWFoL of adolescents and indirectly influences their SWFoL through WLB. These revelations are vital for guiding interventions and support programs to improve family interactions. Through our multi-group analysis, we found that in families where fathers demonstrate low gender-transcendent attitudes, the link between mothers’ perceived partner support and WLB is notably stronger. These results are discussed in detail in the following sections.

### 4.1. Partner Support, Work–Life Balance, and Satisfaction with Food-Related Life

Drawing on the W–HR model [16], Hypothesis 1 posited that perceived partner support positively relates to SWFoL for fathers and mothers. This hypothesis was supported for fathers but not mothers, which aligns with findings reported by Schnettler et al. [13] regarding a positive and direct association between perceived emotional family support and SWFoL in fathers but not mothers. This suggests that family emotional support and partner instrumental and emotional support show similar behavior in dual-earner couples. However, a plausible explanation for the current findings may be related to the lower scores of mothers than fathers regarding perceived partner support. It can be hypothesized that mothers do not receive enough emotional and instrumental support, which they highly value, according to Alarifi and Basahal [24]. This lack of support may hinder their ability to perform well while cooking or to be in a good mood for sharing family meals [16]. In this context, Schnettler et al. [13] suggested that women still bear the primary responsibility for food-related household tasks, even if they are employed. For working mothers, preparing family meals can become a source of stress. If they do not receive instrumental support from their partners in managing their household-related responsibilities, their SWFoL could be impacted.

Our second hypothesis posits that one parent’s perceived partner support is positively linked to the other parent’s (H2a) and the adolescents’ (H2b) satisfaction with food-related life. Hypothesis 2a was not supported; no direct crossover effects were observed between parents’ perceived parent support and SWFoL. These results align with those reported by Schnettler et al. [13] when studying the relationships between emotional family support and SWFoL in dual-earner parents. These findings again suggest that parents’ perceived partner and emotional family support follow a similar behavior in dual-earner parents. One possible explanation may be related to the fact that adolescents also provide emotional support to their parents [51], which has not been evaluated in the present study. By contrast, H2b was partially supported, given that a positive and direct partner effect was found from mothers’ perceived partner support to the adolescents, enhancing their SWFoL. This finding expands on the knowledge, suggesting that although perceived partner support in mothers was lower than in fathers, mothers gained personal resources from the support that fathers gave them, which allowed mothers to effectively allocate personal resources for non-work needs [23], improving their adolescent children’s SWFoL. However, more research is needed to determine whether emotional or instrumental support from fathers explains the positive association between mothers’ perceived partner support and adolescents’ SWFoL.

Based on the COR theory [26], our third hypothesis posits that perceived partner support positively relates to work–life balance for fathers and mothers. This hypothesis was supported, suggesting that both parents’ perceived partner support was positively associated with their WLB. These findings align with those of previous studies assessing the association between emotional family support and WLB in Chile [13,27] and the relationship between partner support and WLB in Korea [29]. In this regard, it has been suggested that spousal support fosters individuals’ feelings of value, acceptance, and care within their partnerships. Therefore, it encompasses providing comfort, appreciation, attention, and assistance from one spouse to another. This support can significantly enhance an individual’s capacity to effectively address and resolve conflicts [11], enhancing their WLB. It should be noted, however, that the differences between the perceived partner support scores between mothers and fathers are associated with a stronger relationship between perceived partner support and WLB in fathers than in mothers. Nevertheless, parents did not differ in their WLB scores; thus, mothers may be using other resources not provided by fathers’ partner support (i.e., from their children or other sources of social and instrumental support, such as their mothers) to achieve a similar level of WLB compared to fathers.

The COR theory also posits the potential transmission of resources between partners via crossover [14]. Based on this, our fourth hypothesis anticipated that one parent’s perceived partner support positively relates to the other parent’s work–life balance. This hypothesis was supported only for fathers, not mothers. An asymmetrical crossover effect was obtained from the fathers’ perceived partner support to the mothers’ WLB, suggesting that only fathers’ perceived partner support increases mothers’ WLB, but not vice versa. This finding is congruent with previous results reported in Chile [13] and Korea [29], respectively, when studying the association between WLB and emotional family and partner support in couples. A possible explanation may be the gender-specific socialization processes [1]. Hence, the asymmetrical crossover effect obtained may arise from women being socialized to be more attuned to relationships compared to their male counterparts [55].

Our fifth hypothesis poses that work–life balance is positively associated with satisfaction with food-related life for fathers and mothers. The findings indicate that H5 was supported for mothers and fathers, aligning with previous results for Chilean dual-earning couples [13]. Thus, it is possible to suggest that resources associated with WLB gained by parents from the perceived partner support also allow them to achieve a higher SWFoL, expanding the knowledge regarding other types of resources that will increase WLB and SWFoL in dual-income couples. These findings might be related to the association between WLB, diet quality [30,31,32], SWFoL, and healthier eating habits (e.g., [35,36]) obtained in different countries. Nevertheless, these findings might also be related to food’s social and hedonic dimensions observed in Chile, as well as in a meta-analytics study [2,3,4,5,6,7,8,9,10,11,12,13,14,15,16,17,18,19,20,21,22,23,24,25,26,27,28,29,30,31,32,33,56]. Regardless of the above, it should be highlighted that the association between mothers’ WLB and SWFoL was of low strength, whereas the same link was of medium strength in fathers.

Based on the W–HR model [16,33], our sixth hypothesis proposes that one parent’s work–life balance is positively associated with the other parent’s (H6a) and the adolescent’s (H6b) satisfaction with food-related life. The findings suggest another asymmetrical partner effect, again from fathers’ WLB to mothers’ SWFoL (H6a partially supported), confirming previous results in Chilean couples [13]. This finding could also be attributed to the socialization of women, which often makes them more attuned to relationships than men [55]. Hypothesis 6b was also partially supported; only fathers’ WLB crossed over to the adolescents, increasing their SWFoL. This finding suggests that only fathers’ WLB allowed them to provide to the adolescents healthier and/or tastier food [3,35,36] or a more significant social experience during family meals [2,3,33], enhancing their SWFoL. Nevertheless, the lack of a partner effect from mothers’ WLB on the SWFoL of fathers and adolescents may be attributed to the notion that when the actor effect is weak, as seen with mothers in this research, the partner effect is probably absent [57]. This plausible explanation suggests that mothers may more frequently use the resources provided by WLB in other activities beyond cooking or enhancing the atmosphere of family meals, such as in other housework chores and caring duties, than fathers, as shown in Table 1, which reveals the higher number of hours per day that mothers spent on housework and caring for their children when compared to that of fathers.

### 4.2. Mediating Role of Family-to-Work Enrichment

Considering both actor and partner effects, our seventh hypothesis assessed the mediating role of WLB in the relationship between parents’ perceived partner support and the three family members’ SWFoL. The findings partially supported this hypothesis, revealing five significant mediating roles—two intraindividual and three interindividual. Both intraindividual mediating roles show that perceived partner support indirectly and positively impacts SWFoL by increasing the WLB in parents. These findings align with the COR theory, which describes a resource gain spiral [17] in which resources gained from perceived partner support generated personal resources (i.e., WLB), which allows for the improvement of SWFoL in parents. These results also align with the results of previous studies reporting that WLB mediates between different types of social support and diverse outcomes, including SWFoL, in other countries [13,27,37,38].

Our findings expand the current understanding of WLB by illustrating its role in mediating relationships among individuals. These important findings reveal that resources gained from perceived partner support can indirectly influence others’ SWFoL through WLB. Specifically, our results suggest two key points: first, that the support fathers perceive from their partners can enhance mothers’ SWFoL by increasing mothers’ WLB; second, that fathers’ perceived partner support can also boost mothers’ SWFoL by enhancing the fathers’ own WLB. Additionally, when fathers feel supported by their partners, it equips them with the necessary WLB to positively affect adolescents’ SWFoL. These results suggest a virtuous cycle: higher perceived support from mothers benefits the fathers and enhances the SWFoL of both mothers and adolescents through improvements in WLB. The practical implications of these findings can empower individuals to improve their well-being and that of others by fostering effective WLB and mutual partner support.

A summary of the outcomes for fathers suggests that they experienced both direct and indirect positive influences on their SWFoL, which are associated with their perceived partner support. Their own WLB mediated the indirect influence. Conversely, mothers primarily experienced indirect effects on their SWFoL stemming from both their partner’s support and the fathers’ partner support, with these influences also mediated through their own, as well as their partners’, WLB. These findings explain that although fathers reported higher levels of perceived partner support than mothers, there were no significant differences in their WLB and SWFoL scores.

Regarding adolescents’ SWFoL, our findings suggest that they received a direct positive effect from their mothers’ perceived partner support (i.e., provided by fathers) and an indirect impact from fathers’ perceived partner support (i.e., supplied by mothers) via their WLB. Hence, although the direct effect was stronger than the indirect effect, support from mothers’ and fathers’ partner support is essential for the adolescents’ SWFoL.

### 4.3. The Impact of Parental Attitudes That Go Beyond Conventional Gender Roles

The results of our last hypothesis suggest that only fathers’ transcendent attitudes influenced one of the relationships examined in hypotheses 1–6. Specifically, the relationship between mothers’ perceived partner support and WLB was more substantial in families where fathers exhibited lower gender-transcendent attitudes. In contrast, this relationship weakened in families where fathers held higher gender-transcendent attitudes. These findings were unexpected, as fathers with low gender-transcendent attitudes (who adhere to more traditional roles) typically contribute less to household duties [20], resulting in lower emotional and instrumental support for mothers in our study. Consequently, mothers had to manage most of the housework, cooking, and caregiving tasks, which theoretically hindered their ability to achieve a better WLB. While this finding warrants further investigation, one possible explanation is that these mothers might seek alternative instrumental support, such as from their adolescent children and through housekeeping assistance.

Regardless of the above, although fathers participate in childcare, housework, and cooking chores in the sample under study, mothers spend 3 h more than fathers on these tasks. These results confirm the gender gap in housework contributions [42].

## 5. Limitations

The limitations of this study require careful consideration. Firstly, the cross-sectional design and non-probabilistic nature of the sample restrict our ability to establish causal relationships and limit the generalizability of the findings. As a result, longitudinal research is needed to clarify causal relationships. The sample comprised only dual-income parents with children aged 10 to 15. Future studies should include families at different life stages to enhance the diversity of the findings. Although the email sent to the families included instructions for each family member to answer the questionnaires separately, it is possible that in some families, mothers or fathers may have been present when the adolescents responded to their questionnaire or vice versa; thus, the answers may have been affected by social desirability. Furthermore, this study used a scale that measures emotional and instrumental partner support, meaning that these two types of support were not assessed separately. Future research should investigate the associations between perceived partner support, WLB, SWFoL, or other domains using a scale that differentiates between emotional and instrumental partner support. This distinction is essential, as previous studies have indicated that instrumental support is often more relevant than emotional support [11]. Finally, it is crucial to recognize that family, cultural, and economic factors significantly impact family dynamics. Thus, conducting cross-cultural studies in this area is essential. There is a particular need for cross-cultural analyses in other Latin American countries and those with varying gender role attitudes, as cultural context largely influences the gendered division of labor, especially regarding food-related responsibilities.

## 6. Conclusions

This study thoroughly examined the direct and indirect influence involving partner and actor effects, focusing on perceived partner support, work–life balance (WLB), and satisfaction with food-related life (SWFoL) among different-sex dual-income families with adolescent children. Additionally, it explored how parents’ gender-transcendent attitudes influence these relationships. Eight hypotheses were posited. Out of these hypotheses, two received full support. Hypothesis 3 (H3) indicated that both parents’ perceived partner support positively influenced their WLB, while Hypothesis 5 (H5) stated that both parents’ WLB positively affected their SWFoL. However, part of Hypothesis 2 (H2) was not supported. Specifically, H2a, which proposed that one parent’s perceived partner support would affect the other parent’s SWFoL, was not supported. The remaining hypotheses received partial support. Hypothesis 1 (H1) was only supported for fathers, indicating that fathers’ perceived partner support positively influenced their SWFoL. Hypothesis 2b (H2b) was supported solely for mothers, suggesting that mothers’ perceived partner support positively affected adolescents’ SWFoL. Hypothesis 4 (H4) was also supported only for fathers, showing that fathers’ perceived partner support positively impacted mothers’ WLB. Hypothesis 6 (H6) was supported exclusively for fathers, indicating that fathers’ WLB positively influenced mothers’ and adolescents’ SWFoL. In Hypothesis 7, five mediating roles of WLB were significant, partially supporting this hypothesis. Furthermore, fathers’ gender-transcendent attitudes moderated only one pathway in the model: the relationship between mothers’ perceived partner support and WLB.

In summary, mothers’ SWFoL was indirectly and positively affected by both their and the fathers’ perceived partner support through both parents’ WLB. Fathers’ SWFoL was directly and positively affected by their perceived partner support and indirectly via their WLB. Adolescents’ SWFoL was directly and positively affected by mothers’ perceived partner support and indirectly by fathers’ perceived partner support through fathers’ WLB. Thus, it can be concluded that the mediating role of work–life balance is significant, as it facilitates the transmission of resources within and between individuals. This transmission occurs from the home environment to the food domain, linking partners’ perceived emotional and instrumental support to increased SWFoL for fathers, mothers, and adolescents. The parents’ gender-transcendent attitudes have scant influence on the results of the present study.

## 7. Implications

Our findings have significant implications for theory, research, and practice. The theoretical implications emphasize the broad applicability of the COR theory—commonly used to examine the work–family interface (e.g., [7,55])—in analyzing the resources gain spiral. This framework helps us understand how perceived partner support and work–life balance enhance satisfaction with food-related life, both at the individual and interindividual levels, demonstrating its versatility for research.

The findings from this study emphasize the essential need for subsequent studies to assess the effects of partner and family support independently. Additionally, future investigations must analyze and differentiate between the impacts of practical and emotional support partners provide in enhancing work–life balance. Moreover, future research should explore whether emotional and practical support contribute directly or indirectly to improved nutrition and the social aspects of satisfaction with food-related life.

From a practical standpoint, our research emphasizes the importance of increasing partner support and promoting work–life balance among dual-earning parents. These improvements can enhance parents’ and adolescents’ satisfaction with their food-related lives, leading to healthier food choices and better family interactions during meals. Therefore, health and labor authorities and organizations should develop initiatives to achieve a more equitable division of household responsibilities among dual-earner families, especially for male employees. Organizational policies have the potential to enhance employees’ work–life balance by implementing measures such as family-friendly workplace initiatives and flexible scheduling options. Additionally, promoting gender equality in the engagement of working parents with their family and personal responsibilities is essential. Such initiatives would enable them to share household duties more fairly with their female partners.

## Figures and Tables

**Figure 1 nutrients-17-01018-f001:**
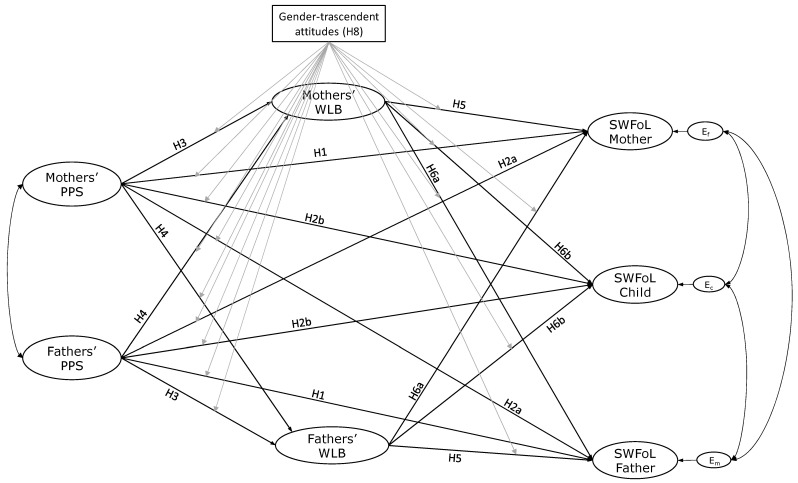
Conceptual model of the proposed actor and partner effects between perceived partner support (PPS), work–life balance (WLB), and satisfaction with food-related life (SWFoL) in dual-earner parents with adolescents. Black lines indicate direct effects between variables, while gray lines represent the moderate effect of parents’ gender-transcendent attitudes. The path diagram does not illustrate the indirect effect of WLB (H7) or control for the influences of each partner’s age, employment type, working hours, family SES, number of children, and frequency of family suppers on the dependent variables (WLB and SWFoL).

**Figure 2 nutrients-17-01018-f002:**
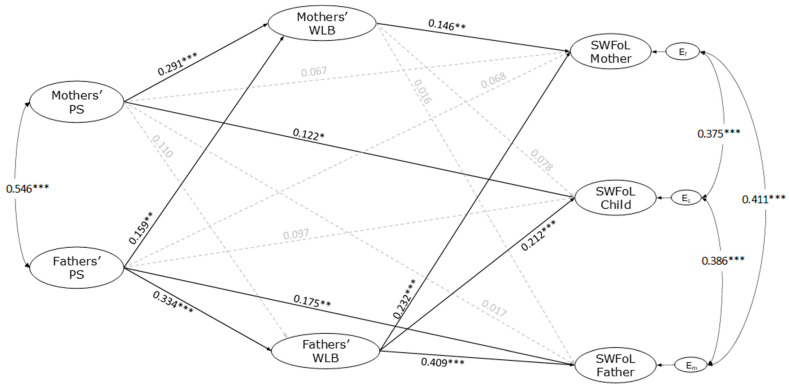
Actor–partner interdependence model of the effect perceived partner support (PPS), work–life balance (WLB), and satisfaction with food-related life (SWFoL) in dual-earner parents with adolescents. E_m_, E_c_, and E_f_: residual errors on SWFoL for mothers, adolescents, and fathers, respectively. * *p* < 0.05, ** *p* < 0.01, *** *p* < 0.001.

**Table 1 nutrients-17-01018-t001:** Sample characteristics (*n* = 516).

Characteristic	Total Sample
Age [Mean (SD)]	
Mothers	37.1 (6.9)
Fathers	39.8 (7.8)
Adolescents	12.2 (1.7)
Adolescents’ gender (%)	
Male	51.0
Female	49.0
Number of family members [Mean (SD)]	4.2 (.9)
Number of children [Mean (SD)]	2.1 (0.9)
Socioeconomic status (%)	
High	4.1
Middle	85.1
Low	10.9
Number of days per week that families ate together [Mean (SD)]	
Breakfast	3.6 (2.4)
Lunch	3.6 (2.2)
Supper	5.5 (2.0)
Dinner	4.7 (2.8)
Number of hours per day during a week that mothers [Mean (SD)]	
Spend on childcare	5.2 (2.1)
Spend on housework	4.1 (2.4)
Spend cooking	2.8 (2.0)
Number of hours per day during a week that fathers [Mean (SD)]	
Spend on childcare	4.0 (2.4)
Spend on housework	3.2 (1.9)
Spend cooking	1.8 (1.9)
Mothers’ type of employment (%)	
Employee	70.3
Self-employed	29.7
Fathers’ type of employment (%)	
Employee	73.6
Self-employed	26.4
Working hours [Mean (SD)]	
Mothers	33.6 (14.9)
Fathers	42.5 (13.0)
Mothers’ place of work (%)	
Remote	3.1
In-person	86.4
Mixed	10.5
Fathers’ place of work (%)	
Remote	1.6
In-person	92.6
Mixed	5.8

**Table 2 nutrients-17-01018-t002:** Descriptive statistics, correlations, factor loading range, omega values, and average extracted variance (AVE) obtained in the mediation actor–partner interdependence model for Perceived Partner Support (PPV), Work–Life Balance (WLB), and Satisfaction with Food-Related Life (SWFoL) scales in dual-earner parents with adolescents.

		M (SD)	Factor Loading Range	Omega	AVE	Correlations						
						1	2	3	4	5	6	7
1. Mothers’ PPS		12.4 (2.5)	0.881–0.945	0.93	0.83	1	0.459 **	0.342 **	0.273 **	0.233 **	0.219 **	0.230 **
2. Fathers’ PPS		13.0 (2.3)	0.840–0.954	0.93	0.82		1	0.312 **	0.368 **	0.217 **	0.322 **	0.229 **
3. Mothers’ WLB		11.2 (2.4)	0.828–0.889	0.89	0.74			1	0.415 **	0.298 **	0.234 **	0.219 **
4. Fathers’ WLB		11.3 (2.4)	0.845–0.925	0.91	0.78				1	0.333 **	0.420 **	0.282 **
5. Mothers’ SWFoL		22.6 (4.7)	0.693–0.824	0.88	0.60					1	0.452 **	0.443 **
6. Fathers’ SWFoL		22.8 (4.8)	0.684–0.908	0.91	0.68						1	0.422 **
7. Adolescents’ SWFoL		24.0 (4.4)	0.719–0.912	0.90	0.66							1

** *p* < 0.001.

**Table 3 nutrients-17-01018-t003:** Summary of hypotheses related to the mediating role of work–life balance (WLB).

Effect	Estimate	*p*-Value	95% CI	Hypothesis	Result
**From mothers’ PPS to mothers’ SWFoL**					
Total	0.135	0.013			
Total Indirect	0.068	0.003			
Mothers’ PPS → mothers’ WLB → mothers’ SWFoL	0.043	0.013	0.009–0.076	H7	Supported
Mothers’ PPS → fathers’ WLB → mothers’ SWFoL	0.025	0.070	−0.002–0.053	H7	Not supported
Direct	0.067	0.213		H1	Not supported
**From fathers’ PPS to fathers’ SWFoL**					
Total	0.318	<0.001			
Total Indirect	0.143	<0.001			
Fathers’ PPS → mothers’ WLB → fathers’ SWFoL	0.003	0.770	−0.014–0.019	H7	Not supported
Fathers’ PPS → fathers’ WLB → fathers’ SWFoL	0.141	<0.001	0.081–0.219	H7	Supported
Direct	0.175	0.002		H1	Supported
**From mothers’ PPS to fathers’ SWFoL**					
Total	0.067	0.271			
Total Indirect	0.050	0.089			
Mothers’ PPS → mothers’ WLB → fathers’ SWFoL	0.005	0.768	−0.026–0.035	H7	Not supported
Mothers’ PPS → fathers’ WLB → fathers’ SWFoL	0.045	0.065	−0.003–0.093	H7	Not supported
Direct	0.017	0.769		H2a	Not Supported
**From fathers’ PPS to mothers’ SWFoL**					
Total	0.171	0.001			
Total Indirect	0.103	<0.001			
Fathers’ PPS → mothers’ WLB → mothers’ SWFoL	0.023	0.035	0.002–0.045	H7	Supported
Fathers’ PPS → fathers’ WLB → mothers’ SWFoL	0.080	<0.001	0.025–0.122	H7	Supported
Direct	0.068	0.230		H2a	Not supported
**From mothers’ PPS to children’s SWFoL**					
Total	0.168	0.003			
Total Indirect	0.046	0.027			
Mothers’ PPS → mothers’ WLB → children’s SWFoL	0.023	0.175	−0.010–0.055	H7	Not supported
Mothers’ PPS → fathers’ WLB → children’s SWFoL	0.023	0.075	−0.002–0.049	H7	Not supported
Direct	0.122	0.035		H2b	Supported
**From fathers’ PPS to children’s SWFoL**					
Total	0.182	0.001			
Total Indirect	0.085	<0.001			
Fathers’ PPS → mothers’ WLB → children’s SWFoL	0.012	0.222	−0.008–0.032	H7	Not supported
Fathers’ PPS → fathers’ WLB → children’s SWFoL	0.073	<0.001	0.032–0.113	H7	Supported
Direct	0.097	0.085		H2b	Not supported

PPS: perceived partner support. SWFoL: satisfaction with food-related life. 95% CI: 95% confidence interval.

## Data Availability

The data and materials are available from the corresponding author upon reasonable request.

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
