# Peer review of "The Mediating Role of Work–Life Balance Between Perceived Partner Support and Satisfaction with Food-Related Life in Dual-Earning Parents and Their Adolescents"

_nutrients, 2025, doi:10.3390/nu17061018_

Round 1

Reviewer 1 Report

Comments and Suggestions for Authors

In many aspects, the text is not very clear, with such an overflow of hypotheses, theories and results. In addition to the main SWFoL scale, three very simple 3-item tools were used. The use of the triad, i.e. the linking of data to mother, father and child(ren), is undoubtedly an advantage. But as I write below the sample was poorly described.

COMMENTS ABOUT INDIVIDUAL PARTS:

Title: The end of the title is not clear. Is it about dual-earning parents with adolescents or dual-earning parents and their adolescent children

Abstract: There is no information on the overall fit of the model. It would be useful to refer to the theoretical basis used (COR, WH-R, APIM). Only in the revised version will it be possible to assess the results and conclusions described in the abstract. For now, the results are rather selective, and the conclusions must be consistent with those in the main text.

Sample: It has been written that at least one adolescent from the family was surveyed. Does this mean that the data were averaged? One child should have been selected according to the criterion developed (middle, oldest?). It should be noted if families were included more than once (means for each child), as this is a big confounder.

Model:  A simplified model is shown without some factors included in the hypotheses, for which significant results were shown even at the abstract level. I am referring here to the Gender Role Attitudes Scale (GRAS). In addition, there is mention of the relationship of WLB and SWFoL with not presented at the graph : age, type of employment, number of working hours, family SES, number of children, and number of suppers per week (here under the table should it be number of shared suppers).

The right side of the model is unclear.  Why the relationship between the SWFoL of the three family members was not directly tested (neither is the corresponding hypothesis). Only a residuals connection was assumed. It was also assumed that there was no relationship between mother and father WLB. This can be at least addressed in the limitations of the study.

The graphical model treats all variables as latent (ellipses). This is in contradiction with the description of the methods, where the construction of summary indices is indicated. It is difficult to assess whether a path or structural model was estimated. In contrast, in other papers by the same author [13] there are clearly defined models with latent variables.

Introduction with hypotheses: The beginning of the introduction looks very good. I very much appreciate the reference in theory and the theoretical justification of each hypothesis. However, H3 and H4 go beyond the nutritional scope of this journal. And on the other hand, there is H7 referring to indirect effects, which seems most important. I suggest shortening parts 1.1 to 1.4 significantly, showing the model supplemented with GRAS and a list of hypotheses, but with a much shorter overall justification. Such a lengthy justification of each hypothesis is like the discussion that follows.  It is very good that the crossover effect is repeatedly referred to in the introduction. I suggest adding the comment that it can be a symmetric or asymmetric effect. Because the demonstration of an asymmetric effect is really an important result.

Description of methods of analysis: I would suggest a concise and more technical approach to this description. Theoretical digressions on APIM distort the perception of this description. This approach could be moved to the introduction.

Missing from the description of methods is a reference to comparisons between groups of respondents in terms of PSS, WLB and SWFoL. A t-test and ANOVA can be inferred from the results. Here in lines 390-391 it is worth adding a df. It will be interesting to see whether the non-parametric tests will confirm the inference, as the values seem skewed towards positive results.

Results and Discussion: The result that H4 was only confirmed for fathers and not for mothers (line 572) is ambiguous as to whether it is a supporting or supported person. In the graph, the FPSS->MWLB pathway is significant, suggesting that women benefit from support by their partner.  

The conclusions also refer to practical implications. Perhaps it is worth changing the title of the subsection? The conclusions are not very concrete. There is too much of an introduction again in the conclusions, no summary of which hypotheses (how many) were confirmed, a conclusion of what influences SWFoL in the three groups.

Author Response

We appreciate all the remarks made by you, which have made it possible to improve the work. Below, we respond to your remarks, indicating the lines where the corrections have been made. All the changes are in red in the new version of the manuscript:

In many aspects, the text is not very clear, with such an overflow of hypotheses, theories and results. In addition to the main SWFoL scale, three very simple 3-item tools were used. The use of the triad, i.e. the linking of data to mother, father and child(ren), is undoubtedly an advantage. But as I write below the sample was poorly described.

R: Thank you for your suggestions on how to improve the manuscript.

COMMENTS ABOUT INDIVIDUAL PARTS:

Title: The end of the title is not clear. Is it about dual-earning parents with adolescents or dual-earning parents and their adolescent children

R: Thank you for your suggestion. The end of the title was changed following your suggestion.

Abstract: There is no information on the overall fit of the model. It would be useful to refer to the theoretical basis used (COR, WH-R, APIM). Only in the revised version will it be possible to assess the results and conclusions described in the abstract. For now, the results are rather selective, and the conclusions must be consistent with those in the main text.

R: Thank you for your suggestions. The model fit (Line 40-41), COR, WH-R, APIM (Lines 32-33) were incorporated in the abstract. The results and conclusion were reworded based on the changes made in the conclusion section (Line 41-50).

Sample: It has been written that at least one adolescent from the family was surveyed. Does this mean that the data were averaged? One child should have been selected according to the criterion developed (middle, oldest?). It should be noted if families were included more than once (means for each child), as this is a big confounder.

R: Thank you for your questions and comments. More information was added regarding the adolescents (one per family) who responded to the survey (Lines 275-281). In addition, lines 401-412 clarified the number of fathers, mothers, and adolescents who make up the sample.

Model:  A simplified model is shown without some factors included in the hypotheses, for which significant results were shown even at the abstract level. I am referring here to the Gender Role Attitudes Scale (GRAS). In addition, there is mention of the relationship of WLB and SWFoL with not presented at the graph : age, type of employment, number of working hours, family SES, number of children, and number of suppers per week (here under the table should it be number of shared suppers).

R: Thank you for your suggestion. In Figure 1 the effect of gender-transcendent attitudes (H8) was added. However, in both figures, we opted not to include the effect of the control variables so as not to overload the figures (Lines 243-246, 484-487). The number of shared suppers is in Table 1.

The right side of the model is unclear.  Why the relationship between the SWFoL of the three family members was not directly tested (neither is the corresponding hypothesis). Only a residuals connection was assumed. It was also assumed that there was no relationship between mother and father WLB. This can be at least addressed in the limitations of the study.

R: Thank you for your questions. According to Kenny et al. (2006), the APIM controls the mutual impact of the independent variable of both parents by looking at the correlations among their perceived partner support. Furthermore, it considers other sources of interdependence by analyzing the correlations between the residual errors of each member's dependent variables, specifically WLB between parents and SWFoL between the three family members [18]. (Line 359-364). For this reason, the model does not provide the correlations between these variables. In any case, the bivariate correlations between the average scores of these variables are found in Table 2.

The graphical model treats all variables as latent (ellipses). This is in contradiction with the description of the methods, where the construction of summary indices is indicated. It is difficult to assess whether a path or structural model was estimated. In contrast, in other papers by the same author [13] there are clearly defined models with latent variables.

R: Thank you for your comments. SEM was performed with latent variables; this issue was incorporated in line 358. The method for calculating the scores of each scale was incorporated because Table 2 shows the average and standard deviation of each measure and the bivariate correlation between them.

Introduction with hypotheses: The beginning of the introduction looks very good. I very much appreciate the reference in theory and the theoretical justification of each hypothesis. However, H3 and H4 go beyond the nutritional scope of this journal. And on the other hand, there is H7 referring to indirect effects, which seems most important. I suggest shortening parts 1.1 to 1.4 significantly, showing the model supplemented with GRAS and a list of hypotheses, but with a much shorter overall justification. Such a lengthy justification of each hypothesis is like the discussion that follows.  It is very good that the crossover effect is repeatedly referred to in the introduction. I suggest adding the comment that it can be a symmetric or asymmetric effect. Because the demonstration of an asymmetric effect is really an important result.

R: Thank you for your comments and suggestions. Parts 1.1 to 1.4 were shortened (Line 142-218). In Figure 1 the effect of gender-transcendent attitudes (H8) was added. The hypotheses were listed at the end of the introduction (Lines 219-218). The definition of symmetrical and asymmetrical crossover effects was added (Lines 99-103).

Description of methods of analysis: I would suggest a concise and more technical approach to this description. Theoretical digressions on APIM distort the perception of this description. This approach could be moved to the introduction.

R: Thank for your suggestion. Accordingly, this part was moved to the Introduction (Line 122-132).

Missing from the description of methods is a reference to comparisons between groups of respondents in terms of PSS, WLB and SWFoL. A t-test and ANOVA can be inferred from the results. Here in lines 390-391 it is worth adding a df. It will be interesting to see whether the non-parametric tests will confirm the inference, as the values seem skewed towards positive results.

R: Thank you for your suggestion. We revised the data analysis and finally used the following analysis: “The statistical comparison of perceived partner support and WLB between mothers and fathers was performed using the Wilcoxon signed-rank non-parametric test (due to the violation of normality). Furthermore, the comparison among the three family members’ SWFoL was analyzed through repeated measures analysis of variance (RM ANOVA) and specific post-hoc comparisons with the Bonferroni correction” (Line 343-348). Results were corrected accordingly (Lines 407-416).

Results and Discussion: The result that H4 was only confirmed for fathers and not for mothers (line 572) is ambiguous as to whether it is a supporting or supported person. In the graph, the FPSS->MWLB pathway is significant, suggesting that women benefit from support by their partner.  

R: Thank you for your comments. We added a phrase clarifying this result (Lines 593-594).

The conclusions also refer to practical implications. Perhaps it is worth changing the title of the subsection? The conclusions are not very concrete. There is too much of an introduction again in the conclusions, no summary of which hypotheses (how many) were confirmed, a conclusion of what influences SWFoL in the three groups.

R: Thank you for your suggestion. The conclusions were changed accordingly (Line 707-732).

Reviewer 2 Report

Comments and Suggestions for Authors

Dear authors

After reviewing your manuscript, it is evident that the study provides valuable insights into the relationships between perceived partner support, work-life balance, and satisfaction with food-related life in dual-earning parents with adolescents.

However, several areas require improvement to enhance the study’s clarity, methodological transparency, and coherence between objectives, results, and discussion.

The manuscript states that participants were selected using non-probabilistic convenience sampling, but further justification is needed regarding how representativeness was ensured within the target population.

Since the study aims to explore family dynamics, providing additional details on demographic diversity, employment settings, and socioeconomic backgrounds would improve the generalizability of findings. Furthermore, the study claims to have reached data saturation, but it does not explicitly describe how this was assessed, which is particularly important in research using dyadic models. The use of structural equation modeling and the Actor-Partner Interdependence Model is well-justified; however, some analytical choices need further clarification.

Your manuscript should include a clearer rationale for selecting specific control variables and provide more details on model fit criteria beyond the CFI, TLI, and RMSEA, such as SRMR or chi-square values.

Additionally, the discussion section sometimes overinterprets findings by making broad generalizations that exceed what can be reasonably concluded from a cross-sectional design. Adjusting these claims and acknowledging the study’s inherent limitations in establishing causality would improve the credibility of the conclusions. While the integration with previous research is generally strong, the discussion would benefit from a more explicit comparison of findings with studies conducted in different cultural contexts, especially considering how family roles and work-life balance dynamics vary internationally.

Some areas of the manuscript also require greater clarity in distinguishing between emotional and instrumental partner support, as the current analysis does not sufficiently separate these aspects despite their potentially distinct effects on work-life balance and satisfaction with food-related life.

Furhtermore, although ethical approval is mentioned, the manuscript does not specify how participant confidentiality was ensured, particularly in the context of dyadic family data. 

Author Response

We appreciate all the remarks made by you, which have made it possible to improve the work. Below, we respond to your remarks, indicating the lines where the corrections have been made. All the changes are in red in the new version of the manuscript:

After reviewing your manuscript, it is evident that the study provides valuable insights into the relationships between perceived partner support, work-life balance, and satisfaction with food-related life in dual-earning parents with adolescents.

R: Thank you very much for your comments.

However, several areas require improvement to enhance the study’s clarity, methodological transparency, and coherence between objectives, results, and discussion.

R: Thank you for your suggestion to improve the manuscript.

The manuscript states that participants were selected using non-probabilistic convenience sampling, but further justification is needed regarding how representativeness was ensured within the target population.

R: Thank you for your suggestion. To ensure representativeness, quotas that reflect the distribution of families across socioeconomic levels (high, medium, and low) were employed (Lines 268-269).

Since the study aims to explore family dynamics, providing additional details on demographic diversity, employment settings, and socioeconomic backgrounds would improve the generalizability of findings. Furthermore, the study claims to have reached data saturation, but it does not explicitly describe how this was assessed, which is particularly important in research using dyadic models.

R: Thank you for your suggestions. A paragraph regarding Chile’s demographic and socioeconomic background was added (Lines 247-257). Thanks for the comment, but we did not use that concept in the manuscript. Assuming that the term "data saturation" refers to sufficient data for robust estimation, we can clarify that the sample size obtained (516 families) is based on the recommendation of Ledermann et al. [44] who point out that to adequately detect mediated pathways among distinguishable dyads, a sample size of 91 dyads and 249 dyads is needed for actor and partner effects, respectively. As can be seen, our sample size is even larger, seeking to maximize the heterogeneity of Chilean families (Lines 270-275).

Ledermann, T., Rudaz, M., Wu, Q., & Cui, M. (2022). Determine power and sample size for the simple and mediation Actor–Partner Interdependence Model. Family Relations: An Interdisciplinary Journal of Applied Family Studies, 71(4), 1452–1469. https://doi.org/10.1111/fare.12644

The use of structural equation modeling and the Actor-Partner Interdependence Model is well-justified; however, some analytical choices need further clarification.

Your manuscript should include a clearer rationale for selecting specific control variables and provide more details on model fit criteria beyond the CFI, TLI, and RMSEA, such as SRMR or chi-square values.

R: Thank you for your suggestions. A justification for selecting each control variable was added (Lines 368-380). SRMR was added to the methods and results sections (Lines 386-388, 425).

Additionally, the discussion section sometimes overinterprets findings by making broad generalizations that exceed what can be reasonably concluded from a cross-sectional design. Adjusting these claims and acknowledging the study’s inherent limitations in establishing causality would improve the credibility of the conclusions. While the integration with previous research is generally strong, the discussion would benefit from a more explicit comparison of findings with studies conducted in different cultural contexts, especially considering how family roles and work-life balance dynamics vary internationally.

R: Thank you very much for your suggestions. The wording of the discussion section was revised to use the word “suggest” in interpreting the results (Lines 566-667). We acknowledged the study’s limitation in establishing causality (Line 683-684). To the best of the authors’ knowledge, there is no available literature in other cultures regarding the associations of the three variables under study to compare our results. Furthermore, available literature on perceived partner support is also limited. In addition, Reviewer 1 asked us to shorten the sections about WLB in the introduction and focus on food-related issues. Lastly, we recognize that family, cultural, and economic factors significantly impact family dynamics. Thus, conducting cross-cultural studies in this area is essential. There is a particular need for cross-cultural analyses in other Latin American countries and those with varying gender role attitudes, as cultural context largely influences the gendered division of labor, especially regarding food-related responsibilities (Lines 697-701). However, countries or the expression "different countries" were added to the discussion when discussing the results.

Some areas of the manuscript also require greater clarity in distinguishing between emotional and instrumental partner support, as the current analysis does not sufficiently separate these aspects despite their potentially distinct effects on work-life balance and satisfaction with food-related life.

R: Thank you for your suggestion. However, we pointed out this issue as one of the study's limitations and suggested future research using a scale that differentiates between emotional and instrumental partner support (Line 691-697).

Furhtermore, although ethical approval is mentioned, the manuscript does not specify how participant confidentiality was ensured, particularly in the context of dyadic family data. 

R: Thank you for your suggestion. The email sent to the families with the links to answer each questionnaire instructed each family member to answer the questionnaires separately (Line 284-285). However, we included this issue as another limitation of the study (Line 688-691).

Reviewer 3 Report

Comments and Suggestions for Authors

paper title - The mediating role of work-life balance between perceived partner support and satisfaction with food-related life in dual-earning parents with adolescents

here are some comments, suggestions, and questions

in the abstract- should note why partner's support is important, more important work-life balance (a simple statement is fine), because this is the key mediator in your study

in the abstract - should note the research locale

in the introduction - should also expand more why the specific research locale, perhaps some statistical figures or numbers about satisfaction ....

hypotheses formulation is adequate

research instruments, could help to provide how data are interpret and the perceived meaning, .... higher WLB means.....

how about the total effect? would suggest to provide a table for the various direct, indirect, and total effects with their corresponding hypothesis and resulting interpretation

could provide more practical implications. what now?

Author Response

We appreciate all the remarks made by you, which have made it possible to improve the work. Below, we respond to your remarks, indicating the lines where the corrections have been made. All the changes are in red in the new version of the manuscript:

here are some comments, suggestions, and questions

in the abstract- should note why partner's support is important, more important work-life balance (a simple statement is fine), because this is the key mediator in your study

R: Thank you for your suggestion. Accordingly, a phrase was added at the beginning of the abstract

in the abstract - should note the research locale

R: Thank you for your comment. The locale was added in the abstract (Lines 37).

in the introduction - should also expand more why the specific research locale, perhaps some statistical figures or numbers about satisfaction ....

R: Thank you for your suggestion. Accordingly, two paragraphs were added at the end of the introduction (Line 247-264).

hypotheses formulation is adequate

R: Thank you for your comment.

research instruments, could help to provide how data are interpret and the perceived meaning, .... higher WLB means.....

R: Thank you for your comment. The interpretation of the perceived partner support, WLB, and SWFoL scales were added (Lines 305-306, 313-314, 332-333).

how about the total effect? would suggest to provide a table for the various direct, indirect, and total effects with their corresponding hypothesis and resulting interpretation

R: Thank you for your question. Table 3 includes this information.

could provide more practical implications. what now?        

R: Thank you for your suggestion. The section with practical implications was expanded (Line 751-754).

Round 2

Reviewer 1 Report

Comments and Suggestions for Authors

With the corrections made, the paper is outstanding in terms of quality of presentation. I would just very much like to check the consistency of the hypotheses with their presentation in Figure 1. Firstly, variant (a) or (b) should appear if a hypothesis (H2 and H6) has been formulated with these points. Secondly, I have a concern that H6 has been marked as H5. There are six pathways on H5 and none on H6 (here rather without notation without a and b).

Author Response

With the corrections made, the paper is outstanding in terms of quality of presentation. I would just very much like to check the consistency of the hypotheses with their presentation in Figure 1. Firstly, variant (a) or (b) should appear if a hypothesis (H2 and H6) has been formulated with these points. Secondly, I have a concern that H6 has been marked as H5. There are six pathways on H5 and none on H6 (here rather without notation without a and b).

R: Thank you very much for carefully revising Figure 1. H2a (in the right part of the Figure) and H2b (in the middle part of the figure) were correct. H5 and H6 were correct.

Reviewer 2 Report

Comments and Suggestions for Authors

...

Comments on the Quality of English Language

.

Author Response

English was revised throughout the manuscript.